# Smart Design of Nanostructures for Boosting Tumor Immunogenicity in Cancer Immunotherapy

**DOI:** 10.3390/pharmaceutics15051427

**Published:** 2023-05-07

**Authors:** Bohan Yin, Wai-Ki Wong, Yip-Ming Ng, Mo Yang, Franco King-Chi Leung, Dexter Siu-Hong Wong

**Affiliations:** 1Department of Biomedical Engineering, The Hong Kong Polytechnic University, Hong Kong 999077, China; bohanyin93@gmail.com (B.Y.);; 2State Key Laboratory for Chemical Biology and Drug Discovery, Department of Applied Biology and Chemical Technology, The Hong Kong Polytechnic University, Hong Kong 999077, China; 3Research Institute for Sports Science and Technology, The Hong Kong Polytechnic University, Hong Kong 999077, China

**Keywords:** cancer immunotherapy, smart responsive materials, controlled release, synergistic therapy

## Abstract

Although tumor immunotherapy has emerged as a promising therapeutic method for oncology, it encounters several limitations, especially concerning low response rates and potential off-targets that elicit side effects. Furthermore, tumor immunogenicity is the critical factor that predicts the success rate of immunotherapy, which can be boosted by the application of nanotechnology. Herein, we introduce the current approach of cancer immunotherapy and its challenges and the general methods to enhance tumor immunogenicity. Importantly, this review highlights the integration of anticancer chemo/immuno-based drugs with multifunctional nanomedicines that possess imaging modality to determine tumor location and can respond to stimuli, such as light, pH, magnetic field, or metabolic changes, to trigger chemotherapy, phototherapy, radiotherapy, or catalytic therapy to upregulate tumor immunogenicity. This promotion rouses immunological memory, such as enhanced immunogenic cell death, promoted maturation of dendritic cells, and activation of tumor-specific T cells against cancer. Finally, we express the related challenges and personal perspectives of bioengineered nanomaterials for future cancer immunotherapy.

## 1. Introduction

### 1.1. Emerging Immunogenic Boosters for Cancer and Their Limitations

Since the discovery of Coley’s toxins derived from streptococcus pyogenes or serratia marcescens for the reduced death rate in patients with tissue and bone sarcoma, cancer immunotherapy (CIT) has emerged as a promising alternative for modern cancer treatments [1]. Importantly, Janes Allison and Tasuku Honjo were awarded the Nobel Prize in Physiology or Medicine 2018 for their discovery of CIT by inhibiting negative immune regulation, known as the immune checkpoint [2,3]. Although a single application of CIT may not be enough to tackle cancer, especially solid tumors, combining CIT with conventional cancer treatment modalities becomes a powerful strategy to protect patients against recurrence and metastasis.

Numerous approaches have been developed to limit tumor growth, such as cancer vaccines, immune checkpoint blockade (ICB), and chimeric antigen receptor (CAR) T cells. Cancer vaccines can be categorized as cell-based, peptide-based, nucleic acid-based, and viral-based vaccines that can stimulate antitumor immunity with the associated tumor antigens that overcome the immune suppression in tumors and induce both humoral immunity and cellular immunity [4]. However, the soluble formulation of cancer vaccines may not be well distributed in lymph nodes for immunogenicity activation without the assistance of a polymer carrier [5]. Furthermore, a single-type antigen vaccine is not enough to eradicate tumors due to diverse tumor subpopulations that require extra time for antigen extraction. Additionally, tumors can gradually develop immunosuppression to limit the treatment efficacy of the vaccine [6,7]. Hence, ICB drugs, such as cytotoxic T lymphocyte antigen 4 (CLTA4), programmed cell death 1 (PD1), and its ligand, PDL1, were invented to inhibit T cell exhaustion, a dysfunction of the T cell state that can also be induced by immunosuppression [8,9]. ICB is based on the concept of immune tolerance suppression of cancer cells with minimal influence on cancer-reactive lymphocytes.

Nevertheless, the therapeutic outcomes of ICB can be challenged by several issues. For instance, the nonspecific distribution of ICB elevates immune-related side effects in other organs [10]. Moreover, tumors can also develop ICB resistance that suppresses the response rate [11]. The low response rate in patients is a great challenge in clinical application [12]. Lastly, CAR T cells were recently approved by the Food and Drug Administration (FDA) for treating leukemia and lymphoma [13]. CAR T cells possess genetically engineered receptors that can recognize and kill specific tumor cells with augmented T cell functions. Unfortunately, CAR T cell therapy is costly and inadequate to eradicate solid tumors [14]. Moreover, manufacturing CAR T cells by viral transduction can pose patients with risks of undesirable effects [15]. These challenges significantly impede the application of immunotherapy against cancers despite the tremendous efforts in this field. In order to overcome these issues, recent advances in smart-designed nanotechnology and nanotherapeutics are the key strategies to improve the overall performance of the immunotherapeutic effect through their functionalities, such as selective accumulation, long circulation time, remote control of drug release, biosensing, and synergistic therapy [16,17,18,19]. The combination of nanotechnology and immunotherapy can create novel approaches and new opportunities for future treatment.

### 1.2. Reprogramming Tumor Immunogenicity for Enhanced Immunotherapy

Before applying nanomaterials to enhance CIT, it is crucial to understand tumor immunogenicity to design a nanoplatform for various purposes. Tumor immunogenicity describes the immune cell infiltration profile in the tumor microenvironment: low tumor immunogenicity (cold tumor) features low tumor antigen presentation and low T cell infiltration, while high tumor immunogenicity (hot tumor) features extensive T cell infiltration and T cell-mediated tumor killing in the tumor microenvironment (Figure 1) [20]. The exclusion of T cells from the tumor microenvironment without infiltration is also defined as low immunogenicity [21]. The efficiency of ICB immunotherapy relies on tumor immunogenicity [20]. Reasonably, the outcome of ICB is limited when tumor-reactive T cells barely infiltrate the tumor microenvironment [20]. Therefore, shifting the cold tumor into a hot tumor is critical to improve the host immune response and potentiate immunotherapy against tumors.

Several noninvasive methods, such as chemotherapy and radiotherapy, and emerging nanoparticle (NP-enhanced therapies, including photothermal therapy (PTT) and photodynamic therapy (PDT), can trigger immunogenic cell death (ICD) that turns the cold tumor into a hot tumor (Figure 2a) [22,23]. For example, Prussian blue NP-mediated PTT triggered ICD features on neuro2a cells with (1) release of adenosine triphosphate (ATP), (2) high-motility group box 1 (HMGB-1), and (3) surface expression of calreticulin (CRT) [23]. Administration of the PTT-treated neuro2a cells provided the vaccination effect in vivo, protecting the animal from re-challenging neuro2a cancer cell inoculation [23]. This implies that host immunity becomes active by the in situ formation of cancer vaccines through ICD.

Antigen-presenting cells (APCs), such as dendritic cells (DCs), can uptake the released tumor antigen from dead tumor cells that initiate the maturation/activation of cytotoxic T cells [22]. Tumor antigen-specific T cells thereafter search and eradicate tumor cells, the immunity cycle repeats, and the host immune system is turned on [24]. Moreover, ICB can be combined with this process to further maintain the therapeutic capacity of T cells [25]. Importantly, the treatment of cold tumors by ICB is a great challenge because it does not have an adaptive immune response [26]. Apart from T cell biology, the tumor microenvironment may involve other types of immune cells such as macrophages, natural killer (NK) cells, and B cells that can also affect the growth of tumors [27,28,29]. Hence, it is ideal to adopt pharmaceutical treatment such as ICB after the induction of ICD to produce a synergistic effect that maximizes therapeutic efficiency and efficacy. This sequential event may require external interferences that first trigger ICD to build up active tumor immunity and subsequently release immuno-booster to enhance cytotoxic or antitumor immune activities. Rationally designed nanoplatforms based on the requirement of the local tumor immunogenicity and combined therapy are a highly sound footing to achieve effective CIT and suppression of recurrence. In this review, we introduce the unique properties of NPs, highlight representative nanoplatforms for different advanced approaches, and express our perspective on nanotechnology and regulations of adaptive immune systems for CIT.

### 1.3. Mechanism of Combined Therapy with ICB

In vitro experiments lack a complete immunity system, circulation, and extracellular cues, where the triggered immunity cycle in an animal has been neglected in the past. Currently, the in vivo synergistic cross-talk between non-immunotherapy and the immune system has been explored. ICB releases the immune inhibitory brake against the tumor, yet the efficiency of ICB alone remains limited in cold tumors [20]. Tumor immunogenicity seems to be an essential requirement to potentiate ICB. As discussed, non-immunotherapy can generate ICD through the indirect activation of the immunity cycle. Combined therapy with ICB has been a popular research topic. We speculate that the immunity cycle would be involved in tumor killing in vivo; the therapeutic treatment and ICB coordinate with the host immunity to generate a positive feedback system to both improve immunogenicity and kill cancer. This speculation leads us to consider the therapeutic sequence involving non-immunotherapy and ICB immunotherapy. Non-immunotherapy, such as photothermal therapy, as a neo-adjuvant first promotes the immunogenicity of cold tumors. After that, the T cell infiltration in the tumor microenvironment enables the intervention of subsequent ICB treatment.

## 2. Smart-Designed Nanobooster for Immunotherapy

### 2.1. Unique Properties of Nanomaterials for Cancer Therapy

Nanomaterials have been widely employed to treat cancer because of their distinctive characteristics, including nanoscale size, massive surface area, advanced physical/optical properties, and modifiable surfaces for various biomedical applications [30,31,32]. These attractive features offer the following aspects to improve antitumor efficiency: (1) large capacity to encapsulate FDA-approved hydrophobic drugs; (2) long retention time in blood by nonfouling polymer passivation [e.g., poly(ethylene) glycol, PEG] [33]; (3) reduction of renal elimination by controlled sizes [34]; (4) modifiable surface by specific moiety to target cancer cells via receptor/ligand interaction [35]; (5) remote control functionality to mediate drug release and manipulation of local biochemical/immune microenvironment of tumors, such as the concentration of reactive oxygen species (ROS), glutathione (GSH), matrix metalloproteinase (MMP) and pH, and immunogenicity (cold to hot tumor) [36,37]. These nanomaterials can be metal-, polymer-, in/organic-, and semiconductor-based for constructing various nanostructures (Figure 2b) [38,39,40]. Two-dimensional nanomaterials also possess attractive properties for various research types, including large surface areas and useful optical/physiochemical properties. Several reviews have comprehensively discussed the latest progress in developing 2D nanomaterials for cancer theranostic applications [41,42]. Hence, we mainly focus on nanoparticle-based therapy in this review. Thus far, the ideal journey of specially constructed NPs for cancer can enter the following sequence: (1) intravenous (i.v.) injection of NPs; (2) passive and active accumulation of NPs in tumors; (3) imaging technique mediated by NPs to identify tumor location at the optimal time point; (4) internal stimuli such as pH and biomolecules or external stimuli such as light and magnetic field to trigger drug release, PTT/PDT, or other possible routes to kill tumor cells and elicit immunogenicity [39,43,44,45]. Hence, a stimuli-responsive all-in-one or multifunctional nanosystem is highly promising for achieving efficient combined therapy for tumor eradication. We have highlighted the representative patents and clinical trials that relate to nanomaterials for boosting cancer immunotherapy in Table 1 and Table 2, respectively. In this session, we review several representative types of multifunctional nanoplatforms for remote control and noninvasive tumor therapy.

### 2.2. Light-Responsive Nanomaterials for Antitumor Therapy

Most optically responsive nanomaterials are sensitive to ultraviolet (UV, 100–400 nm) [46], visible light (400–900 nm) [47,48,49], and the near-infrared (NIR) first (I, 650–950 nm) to second (II, 1000–1700 nm) window [50], of which the tissue penetration depth increases with the increased wavelength and decreased scattering and absorption of light by tissues. NIR-I and II have shown promise as treatment systems. One important advantage is that the intensity and illumination area of NIR lasers can be easily adjusted without affecting biological reactions. This helps to minimize any unwanted side effects [51]. Additionally, NIR has shown negligible tissue autofluorescence and high spatiotemporal resolution compared to visible light fluorescent imaging [52]. Hence, the NIR window is the ideal option to guide combined therapy. Xu et al. developed a semiconducting polymer nanoengager that highly absorbed NIR-II for PTT and its core was coated with the fused membranes of 4T1 tumor and DCs as the damage-associated molecules (DAMPs) and T cell-stimulating factors, respectively (Figure 3a) [53]. The nanosystem achieved several objectives. Firstly, it targeted 4T1 cells using homotypic fusion. Secondly, it induced localized PTT through NIR-II. Thirdly, it promoted DC maturation through DAMPs and TLR. Finally, it primed T cells via CD80/86 and pMHC. This platform was cleverly designed to achieve nanovaccine effects, systemic immune responses, and ICD-induced immune activation all at once. Consequently, the nanoplatform efficiently suppressed primary and distant tumors and prevented metastasis in the lung and liver by creating immunological memory in 4T1-bearing Balb/c mice.

It is challenging to monitor the drug release profile and immune cell activity in combined chemo- and immunotherapy. However, Hao et al. developed a nanosystem that uses NIR-II to enable multiplexed intravital imaging (Figure 3b) [54]. This approach allows for the visualization of the pharmacokinetics and pharmacodynamics of chemodrugs and transplanted immune cells in vivo. This is an improvement over traditional methods, which have been unable to achieve such results. In this study, the authors utilized two types of nanomaterials: Ag_2_Se quantum dots (QDs) and Ag2S QDs. These materials emitted wavelengths of 1350 nm and 1050 nm, respectively, when excited with an 808 nm laser. The Ag_2_Se QDs were passivated with methoxy PEG (mPEG) 2-distearoyl-sn-glycero-3-phosphoethanolamine (DSPE) and loaded with stromal-cell-derived factor-1α (SDF-1α) and the chemodrug doxorubicin (DOX). After being taken up by the MDA-MB-231 tumor model in nude mice, DOX was released from the Ag_2_Se QDs in the low pH of the endosome/lysosome to kill the tumor cells. NK cells were then inoculated, pretreated with Tat-Ag_2_S and released SDF-1α from Ag2Se QDs, which induced a tropism effect that attracted the NK cells to the tumor microenvironment for immunotherapy to eradicate any remaining tumor cells. This sequential treatment effectively suppressed tumor volumes compared to control groups. The authors used NIR-II fluorescence imaging in vivo to track the pharmacokinetics of chemodrugs (at 1350 nm) and the pharmacodynamics of injected immune cells (at 1050 nm), providing a clear picture of the nanomedicine for enhanced imaging and combined therapy in cancer treatment.

Engineering artificial nanoenzymes to emulate real-enzyme activities for antitumor therapy has been a hot research topic, especially for the immunomodulation of tumor-associated immune cells [55]. Wen and colleagues demonstrated Cu_2–x_Te-based NPs that showed catalytic activity toward GSH depletion (mimicking glutathione oxidase) and ROS generation (mimicking peroxidase) (Figure 3c) [56]. In addition, the semiconductor nanostructure of the Cu_2_-xTe NPs efficiently absorbed NIR-II for a photothermal effect (with a photothermal conversion efficiency of PCE = 39.4%). This enhanced the enzymatic activity of the NPs, aiding in the clearance of primary tumors. The smart nanosystem successfully reversed the immunosuppressive tumor microenvironment and improved the ICD effect (e.g., through elevated levels of ATP, CRT, and HMGB-1 to attract DCs) to activate the immune system (e.g., through increased M1 macrophage polarization by elevated levels of proinflammatory cytokines) for the purpose of killing primary and distant metastatic tumors. This study highlights the potential of light-responsive nanoenzymes in cancer immunotherapy.

**Figure 3 pharmaceutics-15-01427-f003:**
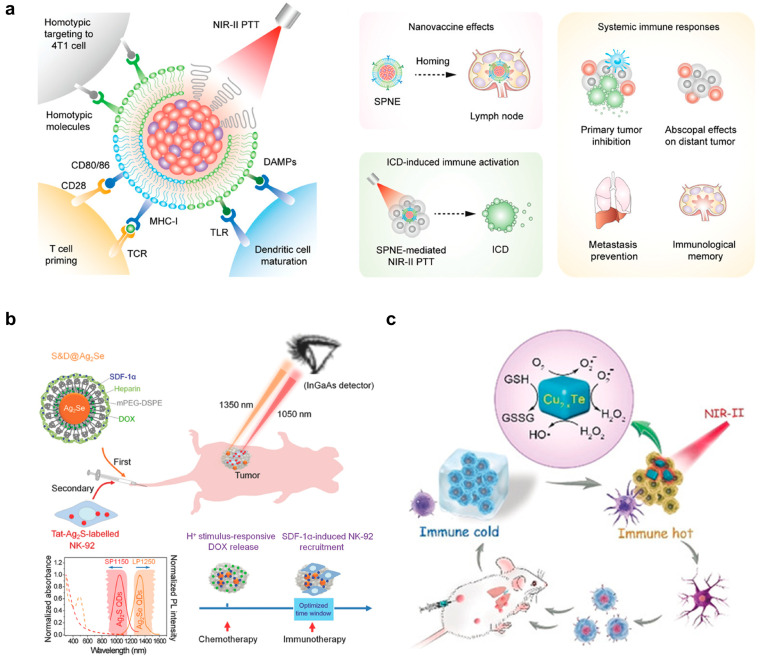
Illustration of light-responsive nanomaterials for tumor therapy. (**a**) Tumor cell- and DC-derived membrane-coated semiconducting polymer-based nanoparticle for tumor cell targeting, NIR-II triggered PTT, DC activation, and T cell priming. The figure was reprinted with permission from ref. [53], copyright of ©2021 Wiley VCH. (**b**) Ag_2_Se and Ag_2_S quantum dots (QDs) for NIR-II multiplexed tumor imaging and chemotherapy, and DC recruitment and tracking for immunotherapy, respectively. The figure was reprinted with permission from ref. [54], copyright of ©2019 American Chemical Society. (**c**) Cu_2−x_Te nanoparticle as an artificial nanoenzyme for catalyzing the generation of reactive oxygen species (ROS) and consumption of glutathione (GSH), together with NIR-II mediated PTT, to kill tumor cells and boost ICD. The figure was reprinted with permission from ref. [56], copyright of ©2019 Wiley VCH.

Plasmonic nanoparticles show unique optical properties that make them exciting tools in nanomedicine [57]. These nanoparticles have emerged as one of the most promising ways to achieve multiple tasks, such as optical imaging, targeted chemotherapy, and localized photothermal therapy (PTT) [58,59]. They can be efficient in tumor theranostics and preventing tumor recurrence [60]. Our team has recently reported a NIR-responsive multilayered gold nanoarchitecture with large mesoporosity (436 m^2^g^−^^1^ and 0.8 cm^3^g^−^^1^) to load DOX (loading efficiency = 34.1%) and high PCE (85.5%), and ultrasensitive surface-enhanced Raman spectroscopy (SERS) detection for tumor theranostics (Figure 4) [16]. SERS imaging has several advantages over fluorescence imaging, including its low cost, high photobleaching resistance, unique vibrational spectrum, and low background. With the α_v_β_3_ integrin-binding moiety Arg-Gly-Asp (RGD), this nanoplatform effectively targeted HeLa cell tumor models both in vitro and in vivo. The optimal therapeutic time was determined by the highest SERS signal during receptor-mediated endocytosis. A single 5 min NIR laser illumination (at 0.5 W/cm^2^) at the SERS-identified tumor site induced robust ablation of the tumor through effective photothermal therapy (PTT) and high apoptosis rates of tumor cells due to efficient doxorubicin (DOX) release. There was no cancer recurrence observed after one month of observation, and minimal side effects were observed. Although the immune response was not studied in this treatment, significant tumor cell death may have caused sufficient ICD-related biomolecules to activate immune memory and suppress recurrence. In summary, NIR-based cancer nanotheranostics is a hot trend that combines deep-tissue imaging and synergistic therapy to achieve an effective therapeutic window for cancer.

### 2.3. pH-Responsive Nanomaterial for Antitumor Therapy

The tumor microenvironment features acidity due to the accumulated metabolites without efficient perfusion [61,62]. The pH of the tumor microenvironment is shown in the range of 6.5–6.9, compared to 7.2–7.4 in normal tissue [63,64]. Therefore, this characteristic has become a target for antitumor therapy. For example, Li et al. reported a size-switchable, self-assembled nanoparticle for platinum prodrug delivery to tumors [65]. Briefly, the 80 nm nanoparticle dissociated to 10 nm dendrimers when pH was below 6.8. The small-size dendrimers favored tumor penetration for inhibiting tumor growth in vivo (Figure 5a). To achieve a pH-sensitive release ICB agent, Wang et al. attempted to conjugate antiPD-1 in a methacrylated PEG-based nanoparticle via 3-(bromomethyl)-4-methyl-2,5-furandione, which was a pH-sensitive linker (Figure 5b) [66]. The system also presented choline analog 2-methacryloyloxyethyl phosphorylcholine (MPC) to enhance accessibility across the blood–brain barrier in a glioma-bearing animal model.

Antitumor drugs, such as gemcitabine, paclitaxel, and CBDCA, can induce the expression of PD-L1, which leads to T cell exhaustion via the NF-kB signaling pathway [67]. ICB-mediated immuno-modulation can overcome this critical drawback in combined chemo/immunotherapy [68]. Su et al. reported a nano-micelle with a pH-responsive and MMP-sensitive PEG shell to deliver anti-PD-1 and paclitaxel. MMP-2 and acidity in the tumor microenvironment triggered the degradation of nano-micelle for drug release [68].

Direct activation of T cells through stimulatory receptor binding is an alternative immunotherapeutic strategy besides ICB. APCs prime T cell activation through (1) peptide major histocompatibility complex (pMHC)–T cell receptor binding, (2) CD80/CD86–CD28 binding, and (3) immunostimulatory cytokines such as interleukin-2 (IL-2) [19]. APC-mimetic can be engineered through the surface decoration of CD3 and CD28 under the presence of cytokines for T cell immunomodulation [69]. Importantly, the parameters and the material features of such APC-mimicking materials are easily controlled and characterized, respectively [70]. Additionally, proximal presentation of CD3/CD28 is required to switch on the T cell activation signal [71,72]. To achieve local T cell activation in an acidic tumor microenvironment, Zhang et al. developed an anti-CD3/anti-CD28 modified interlock DNA nanospring, which was featured by pH-responsive, reversible change of the length between 70 nm and 240 nm [73]. The DNA nanospring exhibited a shrunk conformation at pH 6.5, which shortened the spacing of the stimulatory antibodies for T cell activation [73].

Immune cells, such as natural killer cells and lymphocytes, search for the tumor via chemokine-mediated attraction [74]. Therefore, immune cells serve as excellent drug carriers through circulation. For example, Huang et al. developed a potent topoisomerase I poison SN38 nanocapsule to be delivered by T cells [75]. The T cell-delivered nanocapsule achieved prolonged survival in tumor-bearing mice compared to that of the delivery of the free nanocapsule [75]. Meanwhile, the migration of magnetic nanoparticle-labeled cells can be directed by a magnetic field [76,77]. Nie et al. designed an anti-PD-1 conjugated magnetic nanocluster through a pH-responsive benzoic–imine bond to select PD-1^+^ T cells [78]. Thus, PD-1+ T cells were magnetically attracted to the tumor, while the pH-responsive linker was cleaved in an acidic tumor microenvironment for anti-PD-1 release as an ICB immunotherapy [78]. For chemo/photothermal therapy, Zhang et al. reported the delivery of photothermal agent IR-820 and polydopamine@chitosan-doxorubicin nanoparticles by platelets [79]. DOX was released through laser irradiation and the acidity in the tumor microenvironment [79]. In the context of the stimuli-responsive system, immune cells can be bioengineered into a biocompatible and tumor-sensitive therapeutic carrier for multimodal antitumor therapy, besides its immunological function.

### 2.4. Nanocatalytic Activities for Tumor Therapy

Acidity confers tumor microenvironment immuno-suppressive and drug-resistant properties, which hurdle the immune response and chemotherapy, respectively [80,81,82]. The reverse of the acidic microenvironment through metabolic modulation could reduce the immunosuppressive effect, subsequently enabling tumor killing by the immune system.

Lactic acid (LA) is the product of aerobic glycolysis [83]. Accumulation of LA in the tumor microenvironment due to extensive metabolism contributes to immunosuppressive properties, such as impairing effector T cell proliferation and promoting the PD-1 expression of regulatory T cells [84,85,86]. Therefore, inhibiting lactic acid production can rescue immunogenicity. For example, the delivery of siRNA against lactate dehydrogenase A (LDHA) via vesicular lipid nanoparticles neutralized the tumor acidity through the reduction of lactate production that promoted the functionality of tumor-infiltrating CD8+ T cells and NK cells [87]. This neutralization also benefited the efficacy of ICB treatment by anti-PD-1 [87]. In a separate study, the inhibition of tumor metabolism through the nanoparticle-delivered dichloroacetate (DCA) prodrug, a pyruvate dehydrogenase kinase 1 inhibitor, also enhanced the anti-PD-1 ICB treatment [88]. Furthermore, the NP-delivered lonidamine and syrosingopine reduced lactate production and efflux, reduced T_reg_ cells and increased NK cell infiltration in the tumor microenvironment [89]. Taken together, the inhibition of tumor metabolism would relieve immunosuppression and enable tumor-reactive immune cell infiltration. Furthermore, these support the concept that tumor immunogenicity correlates with ICB efficiency in the presence of respective inhibitory ligands.

Glucose is a fuel for metabolism. Starvation therapy aims to control tumor growth through the deprivation of nutrients, such as glucose. Glucose oxidase (GOx) has been a focused enzyme in this field. GOx can convert glucose to gluconic acid and generate H_2_O_2_ for starvation therapy and ROS-mediated killing [90,91]. Since regular metabolism should be maintained in normal tissue, accurate delivery of GOx to the tumor microenvironment is important to prevent systemic toxicity. Zhao et al. demonstrated that GOx-conjugated nano-gel achieved enhanced intratumoral retention and therapeutic outcome compared to free GOx in vivo (Figure 5a) [92]. Moreover, Zhang et al. attempted to deliver a hypoxia-activated prodrug, banoxantrone dihydrochloride (AQ4N), with GOx through separate liposome carriers [93]. The hypoxia condition generated by GOx converted AQ4N from a non-cytotoxic form to a cytotoxic form, synergized with the starvation effect to boost the anti-tumor outcome [93]. As lactate is a source of the tricarboxylic acid (TCA) cycle for energy production [94], Yu et al. reported the co-delivery of GOx and α-cyano-4-hydroxycinnamate (CHC) through zeolitic imidazolate framework-8 [95]. CHC could inhibit monocarboxylate transporter 1 for lactate influx, blocking both lactate and glucose supply enhanced starvation therapeutic efficiency in vivo (Figure 6b).

### 2.5. Magnetic-Responsive Nanomaterials for Remote Control Antitumor Therapy

FDA-approved magnetic materials such as maghemite (γ-Fe_2_O_3_) and magnetite (Fe_3_O_4_) with superparamagnetic properties (typically named superparamagnetic iron oxide nanoparticles (SPIONs)) have been employed for diagnosis (e.g., magnetic resonance imaging and MRI) and therapeutic (e.g., magnetic hyperthermia (MH) by alternating magnetic field (AMF)) purposes [96]. Modern manifestations leverage SPIONs to target tumors by the magnetic field and generate MH for several treatments, including its emerging use in immunotherapy [97]. For instance, Guo and colleagues fabricated a magnetic-responsive nanoagent consisting of SPIONs caged by cationic copolymers, poly(lactic-co-glycolic acid), and mPEG that electrostatically attached cytosine–phosphate–guanine oligodeoxynucleotides (CpG ODNs), an immunoadjuvant, to promote the maturation of DCs via activating TLR9 (Figure 6a) [98]. The complex had high absorption peaks in the NIR wavelength range, making it suitable for NIR-mediated PTT. Additionally, the complex acted as a contrast agent for bimodal imaging using photoacoustic (PA) and magnetic resonance imaging (MRI) to localize the tumor location after the tumor had taken up the complex. The authors used a magnetic field to guide the complex towards the tumor and laser excitation at 663 nm to initiate PTT, inducing ICD as an autologous cancer vaccine. This promoted the activation of DCs/T cells for subsequent immunotherapy, which effectively suppressed the growth and recurrence of metastatic tumors in 4T1-bearing Balb/c mice.

Magnetic manipulation of DC maturation and tracking during the journey of the immune response has become an active area of research as the DC homing rate to lymphoid tissue is poor [99]. Jin and colleagues employed a magnetic-responsive platform that glows to effectively transport a combination of indocyanine green (ICG), iron oxide, and ovalbumin antigen (OVA) to dendritic cells (DCs), thereby improving DCs’ activation and migration to lymph nodes through magnetic guidance (Figure 7b) [100]. Following intravenous injection of nanoplatform-treated DCs into C57B/L6 mice, the magnetic field attraction significantly detained the DCs in lymph nodes and their distribution was observed through both MRI and NIR imaging. This augmented migration dramatically amplified the count of dividing T cells and the secretion of proinflammatory cytokines, such as interferon-gamma (IFN-γ). Consequently, this group exhibited almost no tumor regrowth compared to other control groups. This research indicates that magnetic-responsive nanomaterials have enormous potential to overcome the challenge of low DC homing rate for effective antitumor therapy.

Although iron-based nanomaterials can improve tumor specificity and mediate ferroptosis by Fenton reaction (ROS generation by ferrous (Fe^2+^)/ferric (Fe^3+^)), several limitations, such as potential toxicity, high dosage requirements, and low ROS conversion efficiency, cannot be ignored [101,102]. To resolve these issues, Yu et al. formulated a hybrid core–shell vesicle (HCSV) to enhance localized ferroptosis and the associated ICD triggered by a circularly polarized magnetic field (cpMF) under MRI guidance (Figure 7c) [103]. The vesicle center was made up of ascorbic acids (AA), while the outer layer was composed of a PLGA layer that contained iron oxide nanocubes (IONCs). Consequently, the collapse of the NPs due to the cpMF led to the release of AA and IONCs, which underwent a reduction from ferric to ferrous to adjust the Fe^2+^/Fe^3+^ ratio, thereby causing ferroptosis for ICD and responding to magnetic resonance imaging. The excitation of cpMF led to an increase in the number of mature DCs, tumor-infiltrating CD8^+^ and CD4^+^ cells. Additionally, the authors demonstrated that the MRI R2* signal could be used to monitor the cpMR-boosted Fenton reaction in real-time in the TRAMP-C1 tumor model in C57B/L6 mice. This research highlights the potential of noncontact tumor theranostics using magnetic fields for imaging and ferroptosis-based cancer immunotherapy.

**Figure 7 pharmaceutics-15-01427-f007:**
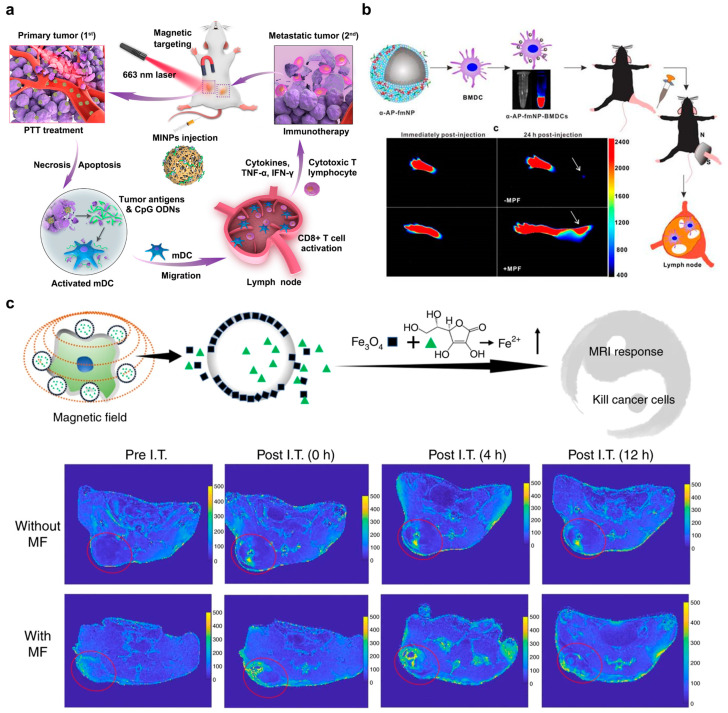
Magnetic responsive nanomaterials for CIT. (**a**) Magnetic/NIR-responsive immunostimulatory nanomaterials in cationic copolymers, poly(lactic-co-glycolic acid), and mPEG that attached cytosine–phosphate–guanine oligodeoxynucleotides (CpG ODNs) to further activate DC after PTT. The figure was reprinted with permission from ref. [98], copyright of ©2019 Elsevier. (**b**) Magnetic labeling of DCs for enhanced lymph node homing rate to promote CIT. The figure was reprinted with permission from ref. [100], copyright of ©2016 Ivyspring. (**c**) Hybrid core–shell vesicle composed of PLGA polymer and ascorbic acids as the shell and iron nanocubes as the cores that can be stimulated by a circularly polarized magnetic field to mediate ferroptosis. Red circles in lower panels indicate tumor sites. The figure was reprinted with permission from ref. [103], copyright of ©2020 Springer Nature.

In short, the above studies showcase the importance of stimuli-responsive nanoplatforms for future simulations of cancer immunotherapy. We summarize the representations of different types of smart design nanoplatforms to simultaneously induce robust ICD and enhance CIT in Table 3.

## 3. Conclusions

In summary, we revisited tumor biology and confirmed that cold tumors should be switched to hot tumors for the immune system to perform “self-governing”. The therapeutic outcomes of conventional therapies and advanced therapies can be reinforced by employing nanomaterials because of their unique properties. In particular, we introduced various state-of-the-art approaches, including light-controlled, pH-responsive, metabolically-modulating, and magnetic-responsive materials to kill tumors and boost adaptive immune responses to fight against recurrence. Specifically, we described the design of nanostructures and the treatment mechanism of each selected example as references for readers who work in this field. Although most nanoplatforms developed by colleagues are still in the fundamental research stage, we believe that more novel and biocompatible nanomaterials will be approved by related authorities, such as the FDA, for future cancer therapy or clinical trials to increase the survival rates of patients with cancer.

## 4. Future Perspectives

In addition to the effectiveness of tumor cell eradication, the biosafety concern of NP-boosted tumor therapy is a hot topic. NP-enhanced tumor immunogenicity generally originates from the dendritic cell uptaking tumor antigens [22,23]. In other words, the tumor antigen from dead cancer cells serves as an in situ cancer vaccine, which activates a highly selective immune response. This mechanism of boosted tumor immunogenicity is similar to the cancer lysate-pulsed dendritic cell vaccination process. Briefly, cancer cell lysate can be generated from ultraviolet B (UVB)irradiation, heating, and freeze–thaw of cancer cells [104]. In a clinical study reported by Tanyi et al., autologous dendritic cells were cultured in vitro and pulsed with cancer cell lysate from the patient to obtain personalized tumor-specific immunity [105]. Most vaccinations (5 to 10 × 10^6^ DCs injected intranodally, 5 doses every 3 weeks) were not associated with grade 3 or higher adverse events, according to National Cancer Institute Common Terminology Criteria for Adverse Events version 4.0 [105]. Although the safety of in vitro cultured autologous dendritic cell-based cancer vaccine has been demonstrated, we believe that standardized and comparative experiments are necessary to search for any potential in vivo immunity impairment or physiological hazards due to different NP-enhanced therapies. The impacts of different therapies, biomaterial designs, and targeted cancer types on immunity might vary. The safety regarding boosted immunogenicity requires further attention in the future.

## Figures and Tables

**Figure 1 pharmaceutics-15-01427-f001:**
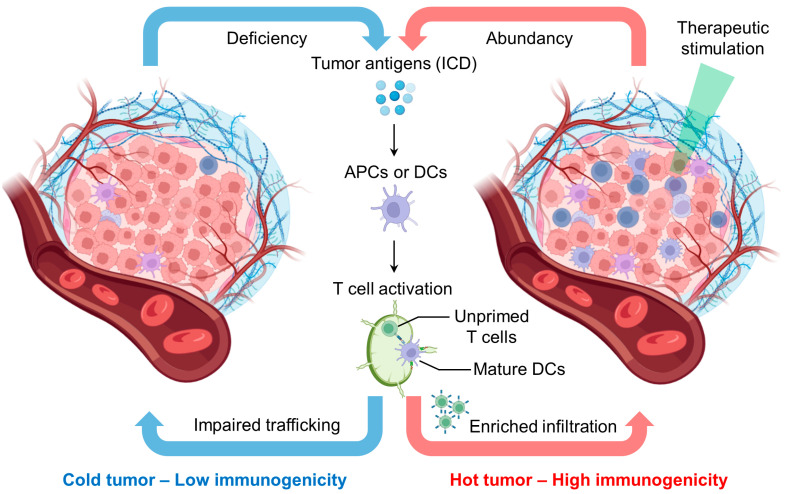
Proposed characteristics of “cold” tumors with low immunogenicity and “hot” tumors with high immunogenicity. “Hot” tumors are often triggered by therapeutic stimulations (e.g., chemo- or radiotherapy) to induce immunogenic cell death (ICD) that releases tumor antigens. The antigens can be uptaken by surrounding dendritic cells (DCs) that undergo maturation and migrate to lymph nodes to activate unprimed T cells. This process enriches the infiltration of tumor-specific T cells and other potential immune cells, such as natural killer (NK) cells and macrophages, to inhibit tumor growth. However, a “cold” tumor is deficient in these features. The image was created by bioRender.com.

**Figure 2 pharmaceutics-15-01427-f002:**
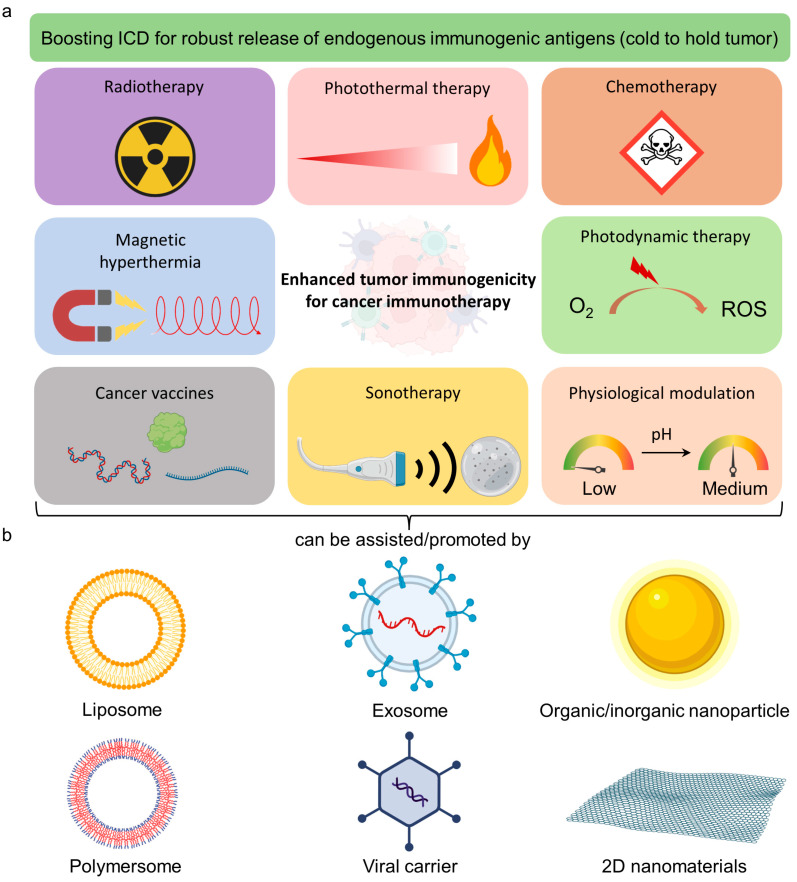
Strategies to enhance tumor immunogenicity. (**a**) Choices of physical, chemical, or pharmaceutical/physiological approaches to alter the whole tumor immunogenicity to boost cancer immunotherapy (CIT). (**b**) The therapeutic outcome of these methods can be further promoted by the applications of multifunctional nanomaterials. The image was created by bioRender.com.

**Figure 4 pharmaceutics-15-01427-f004:**
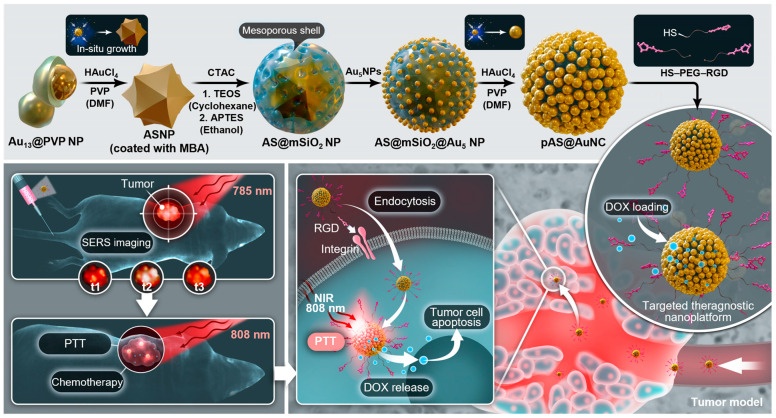
Schematic illustration of surface-enhanced Raman spectroscopy (SERS)-based nanoplatform for tumor theranostics. This nanosystem consists of a multilayered gold nanostructure: gold nanostar/mesoporous silica shell/gold nanocluster that possesses multiple intragap electromagnetic field hot spots for ultraeffective SERS detection and PTT, as well as high mesoporosity for chemodrug loading/unloading under NIR control to achieve robust Raman imaging guided chemo-/photothermal therapy for cancer. The figure was reprinted with permission from ref. [16], copyright of ©2019 Wiley VCH.

**Figure 5 pharmaceutics-15-01427-f005:**
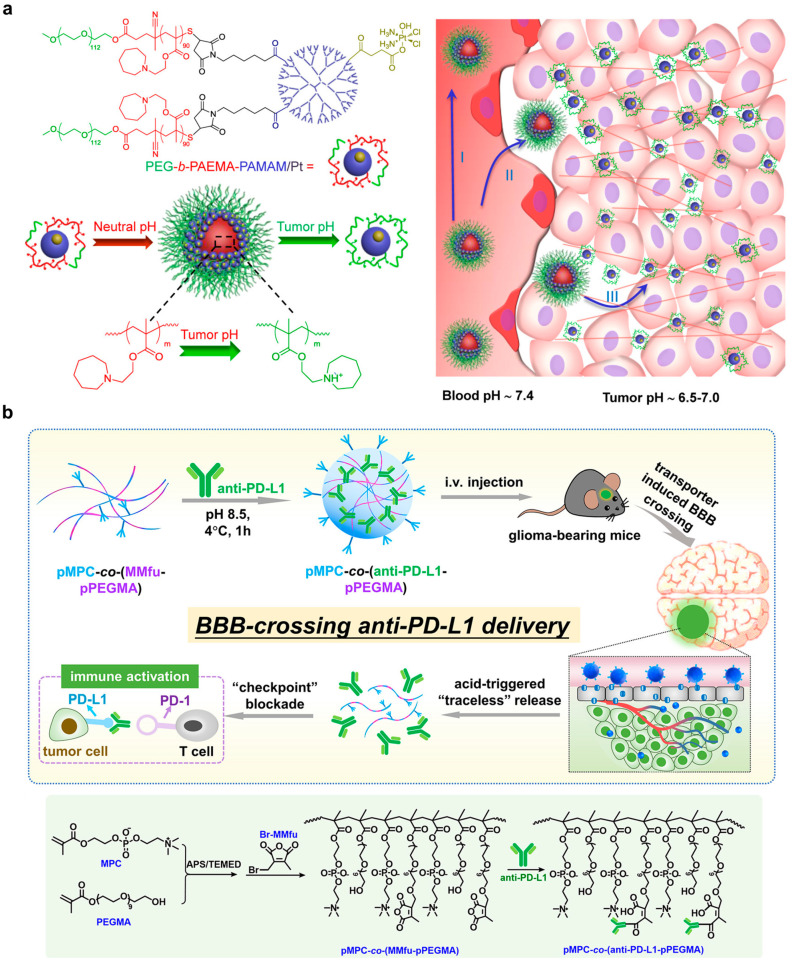
Demonstration of pH-responsive nanomaterials for tumor immunotherapy. (**a**) An amphiphilic polymer-based nanoparticle composed of a polyamidoamine dendrimer shell that can switch its size from ~80 nm to <10 nm in an acidic environment (pH 6.5–7.0) of the tumor to facilitate cellular uptake and deliver platinum-prodrugs. The figure was reprinted with permission from ref. [65], copyright of ©2016 American Chemical Society. (**b**) A blood–brain barrier (BBB)-crossing and pH-responsive nanoplatform based on choline analog 2-methacryloyloxyethyl phosphorylcholine (MPC) that polymerizes with short-chained poly(ethylene glycol) methacrylate (PEGMA) and conjugates with antiPDL1 for ICB therapy. The figure was reprinted with permission from ref. [66], copyright of ©2016 American Chemical Society.

**Figure 6 pharmaceutics-15-01427-f006:**
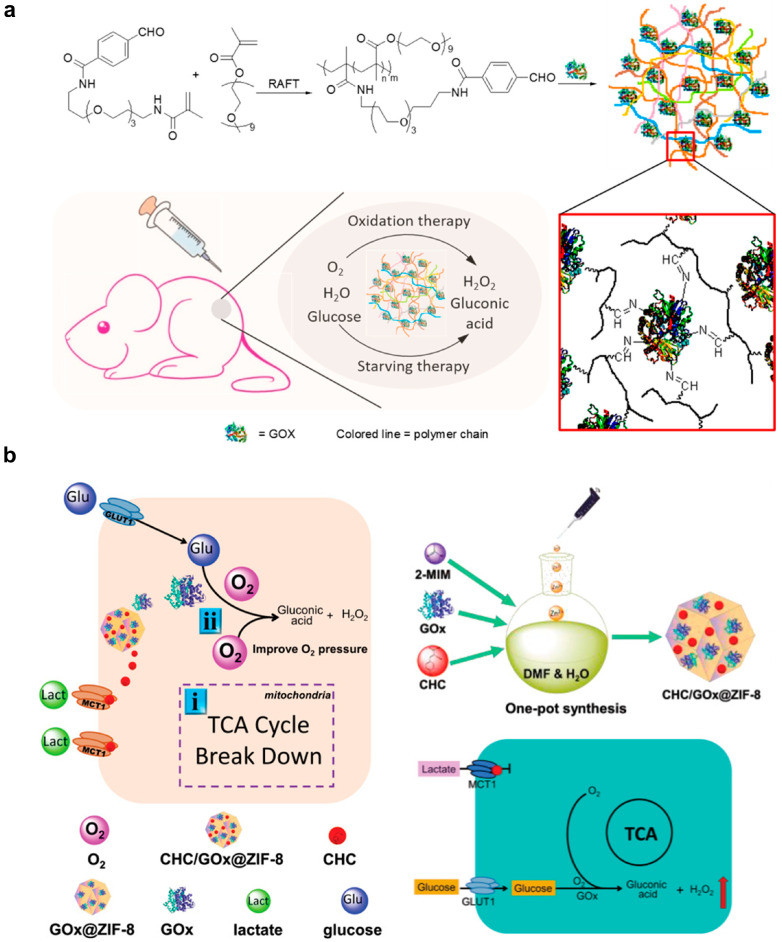
Artificial nanoenzymes manipulate the metabolism of tumor cells to achieve synergistic therapy. (**a**) Glucose oxidase (GOx) incorporated in mesoporous organosilica NPs containing L-arginine for synergistic cancer-starving and nitric oxide therapy. The figure was reprinted with permission from ref. [92], copyright of ©2017 American Chemical Society. (**b**) Zeolitic imidazolate framework-8 (ZIF-8) nanoplatforms for the co-delivery of α-cyano-4-hydroxycinnamate (CHC) and GOx that results in depletion of lactate and glucose, respectively, to strengthen starvation therapy. The figure was reprinted with permission from ref. [95], copyright of ©2021 Wiley VCH.

**Table 1 pharmaceutics-15-01427-t001:** Selected patents related to the application of nanotechnology for enhanced tumor immunotherapy.

Compounds	Claims	Cancer Types	Patent No. and Published Year
Nanoparticles core consisting of metal/semiconductor atoms such as Au, Ag, Cu, Pd, Pt, Gd, and/or Fe and bear specific peptide sequence, FKLQTMVKLFNRIKNNVA and other antigens	At least one adjuvant can stimulate the immune system (T helper cells) response for the prevention and treatment of cancer.	Colon, pancreas, gut, lung, liver, ovary, or bladder cancer	JP2008514686A, 2008
A nanoparticle consists of a polypeptide, FAEKFKEAVKDYFAKFWDGSGLTVSFWYLTVSPWY, with a cholesterol lipid-modified fluorescent dye molecule (DiR-BOA or Fluo-B0A)	The nanoparticle shows synergistic targeting, diagnosis, and treatment of nasopharyngeal cancer by the specific peptide sequence, LTVSPWYLTVSPWY, and fluorescent imaging with enhanced NK cell activity in the tumor environment.	Nasopharyngeal carcinoma	WO2013181934A1, 2013
Zinc phthalocyanine (ZnPcs) and polymers poly(lactic-co-glycolic acid) (PLGA), polyethyleneglycol (PEG), and a lipophilic triphenylphosphonium (TPP) cation nanoparticles loaded with mitochondrial-targeted moiety	This nanoparticle can target TPP moiety in cancer and induce PDT for the production of reactive oxygen species (ROS) under laser stimulation to activate dendritic cells.	Breast cancer	US20150374714A1, 2015
Nanoparticles based on human serum albumin that encapsulates chlorin and catalase and bond with pegylated anti-HER2 nano antibody	The nanoparticle can improve tumor hypoxia and promote immunogenic PDT to treat and inhibit ovarian tumor cells through enhanced immunogenic signals such as damage-associated molecules (DAMPs).	Ovarian cancer	CN113855788A, 2021
Gold nanostars (AuNS) with the anti-PDL1 antibodies	Primary tumors and metastatic cancer sites can be targeted by plasmonics-active (plasmon peak at 600 to 1000 nm) AuNS with a mean tip-to-tip diameter from 10–200 nm and treated with laser-mediated PTT. The co-administration of anti-PDL1 targets the costimulatory molecules (e.g., PD-1 and PD-L1).	Metastatic breast cancer and/or bladder cancer	WO2016209936A1, 2022

**Table 2 pharmaceutics-15-01427-t002:** Recent clinical trials employing nanomaterials for enhanced tumor immunotherapy.

Compounds	Type of Disease	Number, Age and Sex of Participants	Purpose	Primary/Secondary Outcome Measures	Clinicaltrials.gov Identifier and Last Update Year
Ethylcellulose polymer encapsulating Cetuximab and decorated with somatostatin analog	Colorectal Cancer (CRC)	30 adults from 20 years to 60 years with all sexes	To present a novel formulation for targeting and treating CRC safely in patients in a high dose with reduction of side effects to noncancer cells	Establishing pharmacokinetics parameters of Cetuximab in the target cells; determining the bioavailability of Cetuximab after oral and i.v. administration; determining the optimized formulation of Cetuximab	NCT03774680, 2019 (Recruiting)
Nano-scintillator fiber-optic dosimeter	Cancer of the Gastrointestinal, Genitourinary, or Gynecologic Systems	13 adults and older adults of all sexes	To examine the real-time dosimetric monitoring of external beam radiotherapy	Dosimetric accuracy of the device with reference to a commercially available dosimeter; feasibility of clinical application of the nanomaterials for dosimetric monitoring of external beam radiotherapy	NCT02407977, 2018 (Completed)
Quercetin-encapsulated PLGA-PEG nanoparticles	Tongue Squamous Cell Carcinoma (TSCC)	1,000,000 children, adults, and older adults of all sexes	To investigate the anticancer effects of Quercetin, either free or encapsulated by nanoparticles in TSCC cell line	Cytotoxicity, apoptosis, and the gene expression of BCL-2, Bax, and PI3K	NCT05456022, 2022 (Not yet recruiting)
Platinum Acetylacetonate with titania (NPt-Ca)	High-grade, recurrent brain tumor (brainstem glioma) in the central nervous system	8 children (5–14 years old years) of all sexes	To study the enhanced therapeutic effect of NPt-Ca on carriers of the diagnosis of glioma brain stem that shows no response to conventional therapy, including surgery, radiation and chemotherapy	Change in the quality of life; Change in tumor size measured by brain magnetic resonance	NCT03250520, 2023 (Completed)

**Table 3 pharmaceutics-15-01427-t003:** Summary of representative multifunctional/stimuli-responsive nanoplatforms for tumor immunotherapy.

Components	Functions	Therapeutic Outcomes	Dosage	Ref.
Light-controlled materials				
poly(benzobisthiadiazole-alt-thiophene), silicon 2,3-naphthalocyanine bis(trihexylsilyloxide), poly(ethylene glycol)-block-poly(propylene glycol)-block-poly(ethylene glycol), and 4T1 cell- and DC-derived membranes	NIR-II fluorescent imaging; Fused membrane to target 4T1 tumors and activate DCs and T cells; NIR-II photoirradiation to trigger PTT	SPNE directly accumulated in lymph nodes and tumors to exert dual vaccination effects; populations of mature DCs and activated T cells were higher; no recurrence in both primary and distant tumors 30 days post-treatment in 4T1-bearing Balb/c mice	200 µg mL^−1^; 200 µL per mouse	[53]
Ag_2_Se and Ag_2_S QDs, heparin, DOX, mPEG-DSPE and SDF-1α	NIR-II fluorescent imaging; chemo- and immunotherapy; long-term tracking of NK-92 cells; attraction of NK-92 to tumors by chemotaxis	Significantly slowed down the regrowth of MDA-MB-231 tumors in nude mice	1 mg mL^−1^; 200 µL per mouse	[54]
Cu_2–x_Te and DSPE-PEG	NIR-II induced PTT; enzyme-like activities to emulate glutathione oxidase for GSH depletion and peroxidase for ROS generation to kill tumor cells and boost immunomodulation of tumor-associated immune cells	18.6% maturation ratio of DCs; the populations of tumor-infiltrating T helper cells and cytotoxic T lymphocytes were 10- and 11-fold higher than the control group; the growth of distant the tumor was delayed by 64%; the survival rate of mice was over 80% of 4T1-bearing Balb/c mice	2.5 mg kg^−1^	[56]
pH-responsive materials				
poly(ethylene glycol)-*b*-poly(2-azepane ethyl methacrylate)-modified polyamidoamine dendrimer with platinum prodrug conjugation	pH-dependent dissociation for enhanced NP penetration and drug delivery in an acidic environment	Enhanced platinum drug accumulation in BxPC-3 bearing Balb/c nude mice after intravenous injection	40 μg of platinum per mouse bearing a BxPC-3 xenograft tumor	[65]
Choline analogue 2-methacryloyloxyethyl phosphorylcholine presenting poly(ethylene glycol) with conjugation of anti-PD-1 via 3-(bromomethyl)-4-methyl-2,5-furandione	Crossing blood–brain barrier; pH-dependent release of anti-PD-1 for ICB immunotherapy	Promoted antibody accumulation in tumor and survival; enhanced tumor infiltrated CD8^+^ and CD4^+^ T cell proliferation (Ki67^+^); increased sera cytokine level (TNF-α and IFN-γ) in LCPN glioma bearing C57BL/6 mice after intravenous injection	0.8 mg anti-PD-L1 per kg of mice	[66]
Interlocked DNA nanospring, conjugated with anti-CD3 and anti-CD28	Activating T cells in an acidic environment for immunotherapy	Inhibited tumor growth; increased tumor-infiltrating CD8+ T cell population in B16F10-bearing C57BL/6 mice after intratumoral injection	0.2 nmol DNA nanospring per mice; 50 µg BMS-1 (a PD-1/PD-L1 ICB drug) per mice	[73]
Metabolically modulating materials				
Liposome for delivery of lonidamine and syrosingopine	Inhibition of lactate production in tumors for tumor control and immunomodulation	Inhibited tumor growth; prolonged survival; increased infiltrated M1-type macrophage and NK cell; reduced infiltrated M2-type macrophage and T_reg_ cell in 4T1 tumor-bearing Balb/c mice after intravenous injection	2.5 mg lonidamine and 1 mg syrosingopore per kg body weight	[89]
Separate liposomes for delivery of glucose oxidase and banoxantrone dihydrochloride	Glucose starvation, H_2_O_2_ generation and hypoxia-activated prodrug-mediated therapy	Inhibited tumor growth in 4T1 tumor bear balb/c nude mice after intravenous injection	2 mg glucose oxidase and 5 mg banoxantrone dihydrochloride per kg body weight	[93]
Magnetic-responsive materials				
cytosine-phosphate-guanine oligodeoxynucleotides (CpG ODN), superparamagnetic iron oxide nanoparticles, and monomethoxypoly (ethylene glycol)-poly(lactic-co-glycolic acid)-poly-l-lysine (mPEG-PLGA-PLL) triblock copolymers	NIR-I (660 nm) mediated photoacoustic imaging and PTT to guide tumor therapy; contrast agents for MRI; load adjuvant to activate DCs via Toll-like receptor 9	The size of primary and distant tumors decreased with a survival period over 60 days post-treatment; DC maturation level and	5 mg per mouse	[98]
1,2-Dimyristoyl-sn-Glycero-3-Phosphocholine, 1-myristoyl-2-hydroxy-sn-glycero-3-phosphocholine, ICG, OVA peptide, α-AP_gp_100 peptide and superparamagnetic oxide NPs	efficient delivery of indocyanine green/iron oxide/ovalbumin antigen to DCs and enhance the activation and migration efficiency of DCs to lymph nodes under magnetic control	13.2% of the injected DC successfully migrated to lymph nodes by magnetic field compared to 2.6% of the control group; death rates of tumor cells reached 69%; inhibition efficiency of tumor growth was 96%	i.v. injection of 1.2 × 10^6^ DCs treated with the nanoplatforms (16 µg/mL) in 50 µL PBS	[100]

## Data Availability

Not applicable.

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
