# Peer review of "Smart Design of Nanostructures for Boosting Tumor Immunogenicity in Cancer Immunotherapy"

_pharmaceutics, 2023, doi:10.3390/pharmaceutics15051427_

Round 1

Reviewer 1 Report

This work "Smart Design of Nanostructures for Boosting Tumor Immunogenicity in Cancer Immunotherapy" is a very interesting topic and lies under the scope of this journal. However, before final acceptance of the paper following points should be added into the revised manuscript:

1. I request authors to add some patents related to Nanostructures for Boosting Tumor Immunogenicity in Cancer Immunotherapy in a Tabular form.

2. I request authors to add status of clinical trials associated with Nanostructures for Boosting Tumor Immunogenicity in Cancer Immunotherapy in a Tabular form.

3. I request authors to add a separate section in manuscript depicting toxicity and safety concerns of Nanostructures for Boosting Tumor Immunogenicity in Cancer Immunotherapy.

4. I request authors to remove all grammatical and typographical errors present in the manuscript. 

Thanks and regards. 

Author Response

This work "Smart Design of Nanostructures for Boosting Tumor Immunogenicity in Cancer Immunotherapy" is a very interesting topic and lies under the scope of this journal. However, before final acceptance of the paper following points should be added into the revised manuscript:

  1. I request authors to add some patents related to Nanostructures for Boosting Tumor Immunogenicity in Cancer Immunotherapy in a Tabular form.

Response: we have included selected patents in Table 1.

  1. I request authors to add status of clinical trials associated with Nanostructures for Boosting Tumor Immunogenicity in Cancer Immunotherapy in a Tabular form.

Response: we have added several recent clinical trials associated with out topics in Table 2.

  1. I request authors to add a separate section in manuscript depicting toxicity and safety concerns of Nanostructures for Boosting Tumor Immunogenicity in Cancer Immunotherapy.

Response: we thank for the reviewer’s suggestion to include the discussion regarding toxicity and safety. We agree that safety is a significant concern in clinical practice. Since the reports regarding the safety of NP-enhanced therapy boosted tumor immunogenicity remain limited, we added our discussion on future perspective as the following:

Besides the effectiveness of tumor cell eradication, the biosafety concern of NP-boosted tumor therapy is a hot topic. The NP-enhanced tumor immunogenicity generally originates from the dendritic cell uptaking ICD-released tumor antigens. In other words, the tumor antigen from dead cancer cells serves as an in situ cancer vaccine, which activates a highly selective immune response. This mechanism of boosted tumor immunogenicity is similar to cancer lysate-pulsed dendritic cell vaccination process. Briefly, cancer cell lysate can be generated from ultra-violet B (UVB)-irradiation, heating and freeze-thaw of cancer cells. According to a recent clinical study by Tanyi et al. (Science translational medicine 10.436 (2018): eaao5931), most vaccinations (5 to 10 × 106 DCs injected intranodally, 5 doses every 3 weeks) were not associated with grade 3 or higher adverse events, according to National Cancer Institute Common Terminology Criteria for Adverse Events version 4.0. Although the safety of in vitro cultured autologous dendritic cell-based cancer vaccine has been demonstrated, we believe that standardized and comparative experiments are necessary to search for any potential in vivo immunity impairment or physiological hazards due to different NP-enhanced therapies. The impacts of different therapies, biomaterial designs and targeted cancer types on immunity might vary. The safety regarding boosted immunogenicity requires further attention in the future.

  1. I request authors to remove all grammatical and typographical errors present in the manuscript. 

Response: we have revised all grammatical and removed any typos in our manuscript thoroughly.

Reviewer 2 Report

 The manuscript is well written and summarizes the area of immune checkpoints inhibitors and enhancement of the immune system. I would like to suggest to separate between direct therapy and indirect enhancement of the immune system since it is entwined through the manuscript. In addition, I would like suggest to focus the reader on the manuscript's main idea. Furthermore, the images are differ in style, wich is also make it difficult to follow the manuscript. 

Author Response

The manuscript is well written and summarizes the area of immune checkpoints inhibitors and enhancement of the immune system.

I would like to suggest to separate between direct therapy and indirect enhancement of the immune system since it is entwined through the manuscript.

Response: we thank the reviewer for this suggestion. We had initially introduced conventional direct pharmaceutical therapy by the administration of drugs in section 1.1 with their limitations. As explained in section 1.2, the NP-enhanced tumor therapy that causes massive tumor cell death or immunogenic cell death is the general first procedure to turn the tumor into “hot tumor” with high immunogenicity. This direct therapy can provide a noninvasive approach to treating cancer and then stimulate the innate immune memory with the help of subsequent events triggered by those specially designed stimuli-responsive nanomaterials. Hence, direct and indirect therapy are in sequential order with or without the co-administration of immunotherapy drugs. Therefore, the highlighted studies in section 2 assessed the immediate NP-mediated therapeutic effect and the efficiency of immune system activation post-treatment.

In addition, I would like suggest to focus the reader on the manuscript's main idea. Furthermore, the images are differ in style, wich is also make it difficult to follow the manuscript. 

Response: we thank the reviewer for this precious suggestion. We have united all images into a similar arrangement style in our text.

Reviewer 3 Report

The authors of this paper examine the design of smart nanostructures used to improve cancer immunotherapy therapies. Although the work is interesting, some issues need to be reviewed/improved.

Major concerns

1. The introduction should be generic in nature, putting the reader in the context of which forms of cancer/tumors are targeted by immunotherapy approaches, the approximate proportion of treatments tested in the clinic, and which types of cancer it produces the best outcomes in. The introduction in this case is separated into two sections, and it is unclear what the goal of the work is, because nanoparticles (NPs) are not the only nanostructures explored in immunotherapy therapies.

2. The manuscript does not discuss or explain Figure 2. Figure 2A depicts the first section of the introduction (vaccines, radiotherapy, and so on), while Figure 2B summarizes the nanosystems that are commonly utilized as nanovectors in cancer treatments. However, these nanovectors are not further developed in the manuscript, or are only discussed superficially.

3. Gold-decorated nanosystems are becoming increasingly important in phototherapy, with excellent outcomes, and they are barely covered in this review. Figure 4 is not described in the manuscript.

4. Table 1 should be placed prior to the conclusions. And I believe it should be expanded to other nanosystems, such as polymeric nanoparticles.

5. Sections 4.1 and 4.2 should be developed within the manuscript as another section since these techniques have been developed for a few years.

6. The conclusions should be improved by explicitly identifying what this review has contributed.

Minor concerns

1. A list of abbreviations is required due to the enormous number of acronyms used in this manuscript.

Author Response

The authors of this paper examine the design of smart nanostructures used to improve cancer immunotherapy therapies. Although the work is interesting, some issues need to be reviewed/improved.

Major concerns

  1. The introduction should be generic in nature, putting the reader in the context of which forms of cancer/tumors are targeted by immunotherapy approaches, the approximate proportion of treatments tested in the clinic, and which types of cancer it produces the best outcomes in. The introduction in this case is separated into two sections, and it is unclear what the goal of the work is, because nanoparticles (NPs) are not the only nanostructures explored in immunotherapy therapies.

Response: we thank the reviewer for this valuable advice. Our review intended to (Section 1.1) describe the limitations of conventional immunogenic boosters for cancer, (Section 1.2) understand the overview of tumor immunogenicity and (Section 2) highlight representative examples of different stimuli-based nanoplatforms for direct and indirect cancer immunotherapy. Hence, our primary goal aims to discuss the recent advances in nanomaterial-based approaches to turn tumor immunogenicity from “cold” to “hot” with different strategies. We have also expressed our opinions for further advancing the future nano-immunotherapy technology for tumors in terms of (1) integration of ICB with nanoplatforms, (2) engineered cells as the nano/microcarriers for therapeutic drugs and (3) biosafety concerns of nanoparticle-enhanced therapies in Section 4. We agree that nanoparticles may not be the only nanostructures explored in tumor immunotherapy, but they have been frequently explored in other studies. Hence, we have added the following description:

“(5) remote control functionality to mediate drug release and manipulation of local biochemical/immune microenvironment of tumors, such as the concentration of reactive oxygen species (ROS), glutathione (GSH), matrix metalloproteinase (MMP) and pH, and immunogenicity (cold to hot tumor) [37, 38]. These nanomaterials can be metal-, polymer-, in/organic- and semiconductor-based for constructing various nanostructures (Figure 2b) [39-41].”

  1. The manuscript does not discuss or explain Figure 2. Figure 2A depicts the first section of the introduction (vaccines, radiotherapy, and so on), while Figure 2B summarizes the nanosystems that are commonly utilized as nanovectors in cancer treatments. However, these nanovectors are not further developed in the manuscript, or are only discussed superficially.

Response: we thank the reviewer for this reminder. We have actually covered the description of liposome-, polymersome and inorganic/organic-based nanoplatform for immunotherapy in Section 2 and also in the revised Table 1 & 2. For instance, Section 2.4 described that Zhang et al. reported a liposome carrier encapsulating glucose oxidase (GOx) with hypoxia-activated prodrug, banoxantrone dihydrochloride (AQ4N) that starved the tumor as a treatment [Biomaterials 162 (2018) 123-131]. We have laos included the discussion of polymer-based examples in Section 2.2 that Xu et al. developed semiconducting polymer nanoengager for NIR-II mediated PTT and activating the DC/T-cell immune cycle[Adv Mater 33(14) (2021) e2008061]; Section 2.3 also covers the description of pH-sensitive PEG nanoparticle for delivering choline analog 2-methacryloyloxyethyl phosphorylcholine (MPC) for treating glioma-bearing animal model. 2D nanomaterials also possess attractive properties for various research, including large surface areas and useful optical/physiochemical properties. Nevertheless, several reviews have comprehensively discussed the latest progress in developing 2D nanomaterials for cancer theranostic applications [43, 44]. Hence, we mainly focus on nanoparticle-based therapy in this review.

We have also presented our previous work about plasmonic gold nanoparticles for NIR-induced PTT and chemotherapy for synergistic treatment of tumors to suppress tumor recurrence (Figure 4). Nevertheless, nanovectors such as exosomes and viral carriers are often non-stimuli responsive and are not our interest in expanding the discussion.

  1. Gold-decorated nanosystems are becoming increasingly important in phototherapy, with excellent outcomes, and they are barely covered in this review. Figure 4 is not described in the manuscript.

Response: we thank the reviewer for this concern. Figure 4 has actually been described in lines 229 to 245 in our maintext. We have added the following introduction to plasmonic gold nanoparticle properties as the following:

Plasmonic nanoparticles show unique optical properties that make them exciting tools in nanomedicine [56]. These nanoparticles have emerged as one of the most promising ways to achieve multiple tasks, such as optical imaging, targeted chemotherapy, and localized photothermal therapy (PTT) [57, 58]. They can be efficient in tumor theranostics and preventing tumor recurrence [59].

  1. Table 1 should be placed prior to the conclusions. And I believe it should be expanded to other nanosystems, such as polymeric nanoparticles.

Response: we thank the reviewer for this suggestion. We have moved Table 3 before the conclusion. Actually, we have included polymeric nanoparticle-based nanosystems mainly in light-responsive and metabolically modulating materials regions.

  1. Sections 4.1 and 4.2 should be developed within the manuscript as another section since these techniques have been developed for a few years.

Response: we thank the reviewer for these suggestions. We have moved sections 4.1 and 4.2 to sections 1.3 and 2.3, respectively.

  1. The conclusions should be improved by explicitly identifying what this review has contributed.

Response: we thank the reviewer for this advice. We have revised the conclusion as the following:

“In summary, we re-visit tumor biology that cold tumors should be switched to hot tumors for the immune system to perform “self-governing”. The therapeutic outcomes of conventional therapies and advanced therapies can be reinforced by employing nanomaterials because of their unique properties. In particular, we have introduced various state-of-art approaches, including light-controlled, pH-responsive, metabolically modulating and magnetic-responsive materials to kill tumors and boost adaptive immune responses to fight against recurrence. Specifically, we have described the design of nanostructures and the treatment mechanism of each selected example as references for readers who work in this field. Although most nanoplatforms developed by colleagues are still in the fundamental research stage, we believe that more novel and biocompatible nanomaterials will be approved by related authorities such as FDA for future cancer therapy or clinical trial to increase survival rates of patients with cancers.”

Minor concerns

  1. A list of abbreviations is required due to the enormous number of acronyms used in this manuscript.

Response: we have included the list of abbreviations after the perspective section.

Reviewer 4 Report

Dear Authors,

It is advised to overview and refer recent published review articles related with your manuscript. Some examples are indicated below.

1-Nanoparticles as Smart Carriers for Enhanced Cancer Immunotherapy

https://www.frontiersin.org/articles/10.3389/fchem.2020.597806/full

2- Nanomedicine-Boosting Tumor Immunogenicity for Enhanced Immunotherapy

 3-Smart Polymeric Nanoparticles in Cancer Immunotherapy

Pharmaceutics 2023, 15(3), 775; https://doi.org/10.3390/pharmaceutics15030775

 4-Enhancing Cancer Immunotherapy through Nanotechnology-Mediated Tumor Infiltration and Activation of Immune Cells

https://www.ncbi.nlm.nih.gov/pmc/articles/PMC5705528/

Thanks

Author Response

It is advised to overview and refer recent published review articles related with your manuscript. Some examples are indicated below.

1-Nanoparticles as Smart Carriers for Enhanced Cancer Immunotherapy

https://www.frontiersin.org/articles/10.3389/fchem.2020.597806/full

2- Nanomedicine-Boosting Tumor Immunogenicity for Enhanced Immunotherapy

 3-Smart Polymeric Nanoparticles in Cancer Immunotherapy

Pharmaceutics 2023, 15(3), 775; https://doi.org/10.3390/pharmaceutics15030775

 4-Enhancing Cancer Immunotherapy through Nanotechnology-Mediated Tumor Infiltration and Activation of Immune Cells

https://www.ncbi.nlm.nih.gov/pmc/articles/PMC5705528/

Response: we have cited the above references as [40], [17], [38] and [39], respectively. With the following description:

“These nanomaterials can be metal-, polymer-, in/organic- and semiconductor-based for constructing various nanostructures (Figure 2b) [38-40].”

“In order to overcome these issues, recent advances in smart-designed nanotechnology and nanotherapeutics are the key strategies to improve the overall performance of the immunotherapeutic effect through their functionalities, such as selective accumulation, long circulation time, remote control of drug release, biosensing, and synergistic therapy [16-19].”

Round 2

Reviewer 3 Report

The authors have satisfactorily answered all of my questions. The manuscript has now been improved.